# Colonization Ability of *Bacillus subtilis* NCD-2 in Different Crops and Its Effect on Rhizosphere Microorganisms

**DOI:** 10.3390/microorganisms11030776

**Published:** 2023-03-17

**Authors:** Weisong Zhao, Yiyun Ban, Zhenhe Su, Shezeng Li, Xiaomeng Liu, Qinggang Guo, Ping Ma

**Affiliations:** Institute of Plant Protection, Hebei Academy of Agriculture and Forestry Sciences, IPM Innovation Centre of Hebei Province, Key Laboratory of IPM on Crops in Northern Region of North China, Ministry of Agriculture and Rural Affairs of China, Baoding 071000, China; zhaoweisong1985@163.com (W.Z.); 18813168209@163.com (Y.B.); suzhenhe0202@126.com (Z.S.); shezengli@163.com (S.L.); 13315286401@126.com (X.L.)

**Keywords:** *Bacillus subtilis* NCD-2, colonization, growth promotion, rhizosphere microbiome

## Abstract

*Bacillus subtilis* strain NCD-2 is a promising biocontrol agent for soil-borne plant diseases and shows potential for promoting the growth of some crops. The purposes of this study were to analyze the colonization ability of strain NCD-2 in different crops and reveal the plant growth promotion mechanism of strain NCD-2 by rhizosphere microbiome analysis. qRT-PCR was used to determine the populations of strain NCD-2, and microbial communities’ structures were analyzed through amplicon sequencing after application of strain NCD-2. Results demonstrated that strain NCD-2 had a good growth promotion effect on tomato, eggplant and pepper, and it was the most abundant in eggplant rhizosphere soil. There were significantly differences in the types of beneficial microorganisms recruited for different crops after application of strain NCD-2. PICRUSt analysis showed that the relative abundances of functional genes for amino acid transport and metabolism, coenzyme transport and metabolism, lipid transport and metabolism, inorganic ion transport and metabolism, and defense mechanisms were enriched in the rhizospheres of pepper and eggplant more than in the rhizospheres of cotton, tomato and maize after application of strain NCD-2. In summary, the colonization ability of strain NCD-2 for five plants was different. There were differences in microbial communities’ structure in rhizosphere of different plants after application of strain NCD-2. Based on the results obtained in this study, it was concluded that the growth promoting ability of strain NCD-2 were correlated with its colonization quantity and the microbial species it recruited.

## 1. Introduction

Chemical fertilizers have played a vital role in promoting crop growth and increasing crop yield [1]. In intensive cropping system, chemical fertilizers are considered inevitable for obtaining optimum yield of crops. However, overapplied chemical fertilizers have caused serious environmental problems, such as soil salinization and water and soil pollution [2]. Therefore, it is imperative to find alternative methods for increasing crop production. The use of microbial agents is an alternative approach to promote crop growth and reduce the amount of chemical fertilizer [3]. Plant growth-promoting rhizobacteria (PGPR) are biocontrol bacteria that exist in the plant rhizosphere and could promote growth and inhibit disease progression [4]. The mechanisms by which PGPR promote crop growth include promoting the absorption of nitrogen (N), phosphorus (P), potassium (K) and other nutrients by the plants [5], producing plant hormones such as indole-3-acetic acid, cytokinin and gibberellin, inhibiting the growth of phytopathogens in the plant rhizosphere [6], inducing the systematic resistance of plants and increasing their tolerance to abiotic stresses [7] and changing the rhizosphere microbiome structure to increase the population of beneficial microbes and reduce the population of deleterious microbes [8].

Recently, an increasing number of studies have indicated that the rhizosphere microorganisms play an extremely important role in promoting plant growth and increasing plant disease resistance [9,10]. The composition of the rhizosphere microbiome is affected not only by soil physical and chemical properties but also by environmental factors such as drought, salinity and alkalinity [11]. In addition, plant root exudates contain a large number of carbohydrates, which not only supply nutrients for the survival of rhizosphere microorganisms but also serve as signal molecules for regulating the composition of rhizosphere microorganisms [12]. The components of root exudates are plant-dependent, and differences in plants affect the structure of rhizosphere microorganisms [13]. Pii et al. found that the composition of the microbiome was different in the rhizosphere of barley and tomato, as well as in two types of soil, so the soils and crop types jointly regulated the composition of the rhizosphere microbiome [14]. In addition, the composition of the rhizosphere microbiome was related to the plant growth stages, even for the same plant [15]. Since changing the composition of the rhizosphere microbiome is an important mechanism by which PGPR promote crop growth and the composition of the rhizosphere microbiome is shaped by plant types, it is expected that the growth promotion effect of PGPR will be affected by plant species.

*Bacillus subtilis* and closely related species are the most important resources for developing microbial fungicides, partly due to their abilities to produce a variety of antifungal compounds as well as their formation of stress-resistant spores, which are conducive to the development and extension of the shelf life of microbial fungicides [16,17]. Some biocontrol *B. subtilis* strains also promoting crop growth and are characterized as PGPR. *B. subtilis* strain NCD-2 shows typical PGPR characteristics in successfully suppressing soil-borne plant diseases and promote the growth of potato and pepper in field experiments [18,19]. However, the plant growth-promoting effect of this strain was variable when used in different regions and different crops, which became a bottleneck for its extensive application. Microbial agents are known to promote plant growth through dynamics changes in soil microbiomes [20,21,22,23,24]. However, information about the extent to which strain NCD-2 causes the variation of microbiomes in rhizosphere of different crops is little known. The purposes of this study were to analysis the colonization ability of strain NCD-2 in different crops and reveal the plant growth promotion mechanism of strain NCD-2 by rhizosphere microbiome analysis. The results are beneficial for a more scientific and reasonable application of strain NCD-2 in promoting plant growth as well as suppressing soil-borne plant diseases.

## 2. Materials and Methods

### 2.1. Bacterial and Culture Conditions

*B. subtilis* strain NCD-2 was maintained long-term at −80 °C in Luria-Bertani (LB) broth supplemented with 30% glycerol. Before experimentation, strain NCD-2 was revived on solid LB medium at 37 °C overnight and then inoculated in nutrient broth (NB) medium and cultured at 37 °C and 180 rpm for 36 h. Afterward, the fermentation broth was centrifuged at 8000 rpm for 5 min and the cells were collected. The harvested cells were resuspended in sterile distilled water, and the concentration was adjusted to 1.0 × 10^9^ colony forming units (CFUs)/mL for incubation.

### 2.2. Plant Growth Experiment

Seeds of tomato (918 pink tomato cooperative), eggplant (Hangjia No. 1), pepper (Shanyan), cotton (Jimian 11) and maize (Zhengdan 958) were germinated on moist filter paper in darkness at 25 °C. After germination, the tomato, eggplant and pepper seeds were sown into a seedling tray containing sterilized vermiculite and cultured under a light/dark cycle of 16 h and 8 h, respectively. Two weeks later, individual seedlings were transplanted separately into 10 cm diameter pots, each containing a 200 g mixture of soil, vermiculite and peat (2:2:1, *v/v/v*). However, the seeds of cotton and maize were directly planted into pots containing the same mixed substrate. After transplanting, 3 mL of strain NCD-2 cell suspension (10^9^ CFU/mL) was directly added to the root of each seedling, and seedlings inoculated with an equivalent volume of sterile distilled water were used as the control (CK). There were 2 treatments for each crop, 15 plants for each treatment and 3 repetitions. Each plastic pot contained one seedling. The seedlings were grown at 28 °C/20 °C (day/night) with a light/dark cycle of 16 h (600 μmol m^−2^·s^−1^ at 28 °C) and 8 h, respectively, for 5 weeks. The plants were supplemented weekly throughout their whole growth process with 1/2 Hoagland’s nutrient solution.

### 2.3. Assays of Growth Indices

The plant height (the distance from the stem base to growth point) and fresh and dry weights of the plant shoots and roots were recorded at 35 days after inoculation. In brief, the shoot fresh weight (SFW) and root fresh weight (RFW) were measured using an electronic scale after washing them with sterile distilled water and blot-drying them with a paper towel. For shoot dry weight (SDW) and root dry weight (RDW), both the shoots and roots were placed in a drying oven at 80 °C until the weight remained constant [19].

### 2.4. Soil Sample Collection and DNA Extraction

The rhizosphere soils of five seedlings from three biological replicate pots were collected and sieved (2 mm). Briefly, seedlings were dug up from each pot, and their roots were lightly shaken to remove loosely attached soil. Then, the rhizosphere soil was harvested by gently brushing off the soil that was still tightly adhered to the roots. The fresh rhizosphere soil was placed in a sterile tube and frozen at −80 °C for DNA extraction. Soil DNA was extracted by a FastDNA spin kit for soil (MP Biomedicals, Solon, OH, USA). The concentration and purity of the extracted DNA were measured by a Nanodrop 2000 spectrophotometer (Thermo Fisher Scientific Inc., Waltham, MA, USA), and the extracted DNA was used for microbiome analysis and measured the populations of strain NCD-2.

### 2.5. Quantitative Detection of Strain NCD-2 in the Rhizosphere of Five Crops

The colonization abilities of strain NCD-2 in different crop rhizospheres were determined using a real-time PCR system with the NCD-2-specific primers qNCD-F (5′-AGGCAGCATTCAAGCACCAG-3′) and qNCD-R (5′-AGCCAGCGATCATTCCCATC-3′) [25]. Real-time PCR was performed in a 20 μL volume contained with 10 μL 2 × TransStart^®^ Top Green SuperMix (+Dye II), 0.4 μL of each primer, 2 μL DNA and ddH_2_O for a total volume of 20 μL. DNA was amplified with Applied Biosystems QuantStudioTM Real-Time PCR Software (Life Technologies, Foster, CA, USA) under the following PCR conditions: 94 °C predenaturation for 30 s; denaturation at 94 °C for 5 s, annealing at 60 °C for 15 s, and extension for 10 s at 72 °C for 40 cycles. The populations of strain NCD-2 in different crop rhizospheres were compared according to the standard curve built with a 10-fold dilution of the genomic DNA of strain NCD-2.

### 2.6. PCR Amplification, Illumina Sequencing and Data Processing

The bacterial community composition was assessed by sequencing the V3/V4 region of the 16S rRNA genes, which was amplified using the primers 338F (5′-ACTCCTACGGAGGCAGCA-3′) and 806R (5’-GGACTACHVGGTWTCTAAT-3′) [26], and the ITS region of fungal 18S rRNA genes were amplified using the primers ITS1F (5′-CTTGGTCATTTAGAGGAAGTAA-3′) and ITS2R (5′-GCTGCGTTCTTCATCGATGC-3′) [27]. The PCR products from each sample were pooled and visualized on 1.8% agarose gels, purified and sent to Illumina’s MiSeq platform. Illumina MiSeq sequencing was performed at the Majorbio Biopharm Technology Co., Ltd. (Shanghai, China). Processing of the raw sequences was performed using the QIIME 1.9 software. Paired-end reads were assigned to samples based on their unique barcode and merged using FLASH 1.2 software. Reads (average quality score < 20), improper primers and ambiguous bases were discarded before clustering. The effective sequences were clustered into operational taxonomic units (OTUs) at 97% similarity using UPARSE 7.0 software. Soil microbial diversity indices were calculated based on resampled OTU abundance matrices in MOTHUR 1.30 software. ACE index was used to characterize the diversity of the rhizosphere soil microbial communities of different crops. Raw data of 16S rRNA gene and ITS gene obtained from all samples are accessible via NCBI SRA database under accession numbers were PRJNA847255 and PRJNA847257, respectively.

### 2.7. Statistical Analysis

Statistically significant differences in plant height and plant biomass were analyzed by Student’s *t*-test (*p* < 0.05), the population of strain NCD-2 in the plant rhizosphere were evaluated by one-way analysis of variance (ANOVA) using SPSS 17.0. Principal component analysis (PCA) was performed to explore the differences in the soil microbial community composition. Data on the differences in microbial community compositions among treatments were obtained, and the relative abundances of major taxonomic groups at the phylum and genus levels were compared. Phylogenetic Investigation of Communities by Reconstruction of Unobserved States (PICRUSt) software package was used to predict the functional composition of bacterial communities in different samples from amplicon sequencing results. The functional genes were identified from Kyoto Encyclopedia of Genes and Genomes (KEGG) database. FUNGuild database was used to analyze, classify and interpret fungal communities according to fungal functions. The FUNGuild software annotates taxonomic data within the OTU table with corresponding data on its online database, the annotations include the guild, trophic mode and growth morphology; only confidence scores of “Probable” and “Highly Probable” were used. Graphs were generated by Origin 8.0 software (OriginLab, Northampton, MA, USA).

## 3. Results

### 3.1. Effect of Strain NCD-2 on the Biomass for Different Crops

The effects of strain NCD-2 treatment on the plant height of five crops were compared (Figure 1A). The results showed that strain NCD-2 treatment could significantly improve the plant height of tomato, eggplant and pepper, with increases of 19.1%, 15.6% and 26.1%, respectively, but could not significantly improve the plant height of cotton or maize (increases of −10.9% and 1.8%, respectively). Strain NCD-2 treatment significantly increased the biomass of tomato, eggplant and pepper. Compared with CK, SFW, SDW, RFW and RDW of tomato increased by 27.25%, 20.06%, 72.31% and 14.39% after application of strain NCD-2, respectively (Figure 1B). SFW, SDW, RFW and RDW of eggplant increased by 54.32%, 51.63%, 56.22% and 42.93%, respectively (Figure 1C). SFW, SDW, RFW and RDW of pepper increased by 16.01%, 54.23%, 26.50% and 13.52%, respectively (Figure 1D). In contrast, strain NCD-2 treatment only slightly increased the biomass of cotton and maize, and SFW, SDW, RFW and RDW of cotton increased by 1.74%, −0.69%, 25.89% and 49.13%, respectively (Figure 1E). SFW, SDW, RFW and RDW of maize increased by 9.8%, 8.72%, 10.25% and 6.23%, respectively (Figure 1F).

### 3.2. Populations of Strain NCD-2 in Rhizosphere

The DNA concentrations of strain NCD-2 ranged from 100 fg/μL to 10 ng/μL and were used as templates for real-time PCR amplification. A good PCR amplification efficiency (E = 97.5%) was obtained from the standard curve y = −3.392x + 20.577, with a strong correlation (R^2^ > 0.99) between the quantity of B. subtilis strain NCD-2 target DNA (x) and the corresponding Ct (y) values (Figure 2A). The DNA concentration of strain NCD-2 was calculated according to the standard curve, and the population of strain NCD-2 in the rhizosphere soils from different crops was obtained. The results showed that strain NCD-2 had the highest DNA concentration of 24.6 pg/g soil in the eggplant rhizosphere and the lowest DNA concentration of 2.1 pg/g soil in the maize rhizosphere. The DNA concentrations of strain NCD-2 in the rhizospheres of cotton, pepper and tomato were similar at 9.05 pg/g soil, 8.26 pg/g soil and 6.98 pg/g soil, respectively (Figure 2B).

### 3.3. Sequencing Data Summary and Microbial Community Diversity

To understand rhizosphere microbiota structure associated with treatment and non-treatment of strain NCD-2 on different crops, we performed Illumina MiSeq sequencing V3 to V4 region of bacterial 16S rRNA and ITS region of fungal 18S rRNA. For bacterial, 5,866,382 valid sequences (Mean ± SE = 195,546 ± 18,310) were obtained and clustered into 9930 OTUs at 0.97 similarity threshold after quality filtering. The detailed sequencing data of each sample in bacterial are shown in Appendix A. These sequences were classified into 38 phyla, 99 classes, 276 orders, 518 families and 1112 genera. The ACE curves tended to reach plateaus, indicated that the sequencing depth was relatively sufficient in covering the bacterial diversity (Appendix A). For fungi, 3,906,977 valid sequences (Mean ± SE = 130,232 ± 9190) were obtained and clustered into 2837 OTUs at 0.97 similarity threshold after quality filtering. The detailed sequencing data of each sample in fungi are shown in Appendix A. These sequences were classified into 13 phyla, 39 classes, 83 orders, 179 families and 291 genera. The ACE curves also tended to reach plateaus, indicated that the sequencing depth was roughly sufficient in covering the fungal diversity (Appendix A). Soil bacterial and fungal community diversities were calculated at the operational taxonomic unit (OTU) level. For bacteria, compared with CK, there was no significant difference for ACE indices in different crops after application of strain NCD-2. For instance, ACE index was 5287.20 ± 109.10 in eggplant rhizosphere soil after application of strain NCD-2, there was no significant difference in CK treatment (5536.00 ± 357.83). For fungi, ACE index was 648.89 ± 44.84 in tomato rhizosphere after application of strain NCD-2, there was significantly increased in CK treatment (752.42 ± 37.72). Although ACE indices were lower in the rhizospheres of eggplant, pepper and cotton after application of strain NCD-2 than that of CK treatment, there were no significant differences (Table 1).

### 3.4. Effect of Strain NCD-2 on the Microbial Community Structure in the Rhizosphere of Different Crops

Principal component analysis (PCA) results showed that soil microbial community structure was different among five crops, and the microbial community structures of eggplant and maize rhizosphere soils changed after application of strain NCD-2 (Figure 3). The first two principal components in the dataset together explained 73.5% and 71.0% of the total variance in the bacterial (Figure 3A) and fungal (Figure 3B) communities, respectively. In addition, the first principal component (PC1) was the most important, accounting for 57.9% and 42.0% of the total variation of the bacterial and fungal communities, respectively. From the plant species point of view, the bacterial community structures were divided into three groups: the pepper and tomato group, cotton and maize group and eggplant group (Figure 3A), and the fungal community structures were also divided into three groups: the pepper and cotton group, tomato and eggplant group and maize group (Figure 3B). In addition, there were differences in the bacterial and fungal community structures of eggplant soils after application of strain NCD-2. Meanwhile, compared to that in the CK treatment, the fungal community structure of maize changed. However, there were no differences in the bacterial and fungal community structures of the other crops after application of strain NCD-2.

### 3.5. Effect of Strain NCD-2 on the Bacterial Taxonomic Community Composition in the Rhizosphere of Different Crops

The bacterial community composition in the rhizospheres of tomato, eggplant, maize, cotton and pepper were analyzed at the phylum and genus levels by microbiome analysis. Venn analysis showed that 21 core phyla were shared by all groups, and only one phylum (Margulisbacteria) was observed in group of eggplant without treatment of strain NCD-2 (Appendix A). Proteobacteria (44.45–52.34%), Actinobacteria (14.32–20.44%), Acidobacteria (7.4–11.81%), Bacteroidetes (4.88–7.67%), Gemmatimonadetes (3.33–6.75%), Chloroflexi (3.09–6.47%), Patescibacteria (1.43–3.44%), Firmicutes (0.76–2.82%), Planctomycetes (0.37–0.88%) and Cyanobacteria (0.33–1.34%) were the top ten predominant phyla classified in all groups, accounting for more than 95% of all the bacterial sequences (Figure 4A). After application of strain NCD-2, the relative abundances of Proteobacteria and Firmicutes in tomato, eggplant and pepper rhizospheres were more abundant than CK treatment, while the relative abundances of Actinobacteria, Acidobacteria and Cyanobacteria were decreased in eggplant and pepper. The relative abundances of Bacteroidetes and Gemmatimonadetes in cotton and maize rhizosphere were more abundant than CK treatment, while Patescibacteria and Cyanobacteria were decreased. Furthermore, analysis of variance showed that the differences in the mean proportions of Proteobacteria, Actinobacteria, Acidobacteria, Bacteroidetes, Gemmatimonadetes, Chloroflexi, Patescibacteria, Firmicutes and Cyanobacteria in different crops were significant (Figure 4B). At the genus level, there were significant differences in dominant genera abundances of different crops between CK and NCD-2 treatments based on Student’s *t*-test (*p* < 0.05) (Appendix A). *Bacillus* and *Sphingobium* were significantly higher abundances in the rhizospheres of tomato, eggplant and pepper under application of strain NCD-2 than that of CK treatment, only *Lysobacter* was significantly higher abundance in the rhizospheres of eggplant and pepper, *Ramlibacter* was significantly higher abundance in the rhizospheres of tomato and eggplant. *Lysobacter*, *Bdellovibrio*, *Pseudomonas*, *Peredibacter*, *Limnobacter* and *Pedobacter* were more abundances in the rhizosphere of cotton after application of strain NCD-2, while lower abundances of MND1, *Kribbella*, *Rhodococcus* and *Actinoplanes*. *Azotobacter* and *Conexibacter* were more abundances in the rhizosphere of maize after application of strain NCD-2, while *Lysobacter*, *Caulobacter*, *Brevundimonas*, *Acidovorax*, *Kaistia*, *Pseudoxanthomonas*, *Kribbella* and *Sphingobium* were lower abundances.

### 3.6. Effect of Strain NCD-2 on the Fungal Taxonomic Community Composition in the Rhizosphere of Different Crops

The fungal taxonomic distribution at the phylum level is shown in Figure 5. Ascomycota (63.17–78.51%), Mucoromycota (0.02–19.59%), Mortierellomycota (0.69–3.87%), Basidiomycota (0.60–1.60%) and Chytridiomycota (0.07–1.82%) were the top five most abundant fungal phyla classified in the samples of different crops, accounting for more than 80% of all the fungal sequences (Figure 5A). Ascomycota and Basidiomycota were more abundant in the rhizospheres of different crops after application of strain NCD-2 than CK treatment. Analysis of variance showed that the differences in the mean proportions of Ascomycota, Mucoromycota, Mortierellomycota in different crops were significant (Figure 5B). At the genus level, there were significant differences in dominant genera abundances of different crops between NCD-2 and CK treatments based on Student’s *t*-test (*p* < 0.05) (Appendix A). For tomato, *Fusarium*, *Clonostachys* and *Neurospora* in the rhizosphere were significantly higher abundances after application of strain NCD-2 than CK treatment, while the lower abundances of *Byssochlamys*, *Aspergillus* and *Fibulochlamys*. For eggplant, the higher abundances of *Byssochlamys*, *Penicillium*, *Mortierella* and *Cladorrhinum* in the rhizosphere after application of strain NCD-2, while *Fibulochlamys* was significantly decreased. For pepper, *Byssochlamys*, *Acremonium* and *Peziza* in the rhizosphere were significantly higher abundances after application of strain NCD-2, while lower abundances of *Verticillium*, *Nectria* and *Guehomyces*. For cotton, the abundance of *Acremonium* and *Guehomyces* in the rhizosphere were significantly increased after application of strain NCD-2, while *Byssochlamys*, *Neurospora* and *Arthrobotrys* were decreased. For maize, *Byssochlamys*, *Aspergillus*, *Trichoderma*, *Cephalotrichum* and *Neurospora* in the rhizosphere were higher abundances after application of strain NCD-2, while lower abundances of *Arthrobotrys*, *Peziza* and *Simplicillium*.

### 3.7. Predicted Functional Analysis of the Microbial Community

In this study, 23 observed predicted functions were found in the study (Figure 6A). Compared with CK treatment, 4, 18, 22, 7 and 1 clusters of orthologous groups (COG) enriched functions were obtained in tomato, eggplant, pepper, cotton and maize rhizospheres after application of NCD-2, respectively (Appendix A). For eggplant and pepper, among the enriched COG functions included amino acid transport and metabolism (E), nucleotide transport and metabolism (F), coenzyme transport and metabolism (H), lipid transport and metabolism (I) and defense mechanisms (V). For tomato, lipid transport and metabolism (I), transcription (K), secondary metabolites biosynthesis, transport and catabolism (Q), intracellular trafficking, secretion and vesicular transport (U). For cotton, RNA processing and modification (A), chromatin structure and dynamics (B), cell cycle control, cell division, chromosome partitioning (D), translation, ribosomal structure and biogenesis (J), cell wall/membrane/envelope biogenesis (M) and cell motility (N). 

The fungal community of the different treatments was categorized based on the ecological guild and form of nutrition (Figure 6B, Appendix A). At the ecological guild level, the relative abundances of animal pathogen–dung saprotroph–endophyte–epiphyte–plant saprotroph–wood saprotroph enriched in rhizospheres of tomato, pepper, cotton and eggplant after application of strain NCD-2, the relative abundances of animal pathogen–endophyte–fungal parasite–plant pathogen–wood saprotroph enriched in rhizospheres of tomato, pepper, cotton and maize after application of strain NCD-2, the relative abundances of animal pathogen–endophyte–lichen parasite–plant pathogen–soil saprotroph–wood saprotroph enriched in rhizospheres of tomato and pepper after application of strain NCD-2, the relative abundances of animal pathogen–endophyte–lichen parasite–plant pathogen–wood saprotroph enriched in rhizospheres of tomato, maize, cotton and eggplant after application of strain NCD-2, the relative abundances of plant pathogen decreased in rhizospheres of eggplant and pepper after application of strain NCD-2, the relative abundances of dung saprotroph–soil saprotroph–wood saprotrop decreased in rhizospheres of eggplant, maize and tomato after application of strain NCD-2 (Figure 6B).

According to the trophic mode, the fungi community was divided into eight trophic mode groups by the FUNGuild database. The fungal community was screened to 539 identifiable species, which included saprotroph (46.38%), pathotroph–saprotroph–symbiotroph (17.81%), pathotroph (10.02%), pathotroph–saprotroph (8.91%), saprotroph–symbiotroph (7.24%), symbiotroph (6.68%), pathogen–saprotroph–symbiotroph (1.48%) and pathotroph–symbiotroph (1.48%) representing the general abundance of each nutrition method in the identified community. Compared with CK, the relative abundances of functional gene of pathogen–saprotroph–symbiotroph were increased and ranked as pepper (156.62%), cotton (162.38%), tomato (693.83%) and maize (12.10%) after application of strain NCD-2, while they decreased by 84.34% in rhizosphere of eggplant. The relative abundances of functional gene of pathotroph were increased by 119.10%, 22.75% and 11.89% in rhizospheres of eggplant, cotton and maize after application of strain NCD-2. The relative abundances of pathotroph–saprotroph were increased by 13.36% and 58.66% in rhizospheres of pepper and tomato after application of strain NCD-2. The relative abundances of pathotroph–saprotroph–symbiotroph were increased by 35.73% and 13.63% in rhizospheres of tomato and cotton after application of strain NCD-2. The relative abundances of pathotroph–symbiotroph were increased by 126.55%, 51.34% and 7.89% in rhizospheres of eggplant, cotton and tomato after application of strain NCD-2. The relative abundances of saprotroph were increased by 27.37%, 16.63% and 12.51% in rhizospheres of eggplant, pepper and maize after application of strain NCD-2. The relative abundances of saprotroph–symbiotroph were increased by 130.58% and 5.48% in rhizospheres of eggplant and cotton after application of strain NCD-2. The relative abundances of symbiotroph were increased by 55.68% and 39.35% in rhizospheres of pepper and eggplant after application of strain NCD-2 (Appendix A). 

## 4. Discussion

### 4.1. The Colonization and Growth-Promoting Ability of Strain NCD-2

*B. subtilis* strain NCD-2 is a promising microbial agent for soil-borne plant diseases in many crops [18,19,25]. The mechanisms of strain NCD-2 against plant soil-borne diseases and salt resistance has been clarified previously by molecular genetics method [28,29,30,31]; however, the effect of strain NCD-2 on colonization and microbial community in rhizosphere of different crops are relatively lacking. Biocontrol and plant growth promoting effects are positively associated with rhizosphere colonization ability of microbial agent [22,28,30,32]. For instance, the colonization pattern of *Bacillus subtilis* N11, which forms biofilms along the elongation and differentiation zones of banana roots, could be linked to the mechanism of growth-promoting effect and biological control [32]. In this study, strain NCD-2 could significantly promote the growth of tomato, eggplant and pepper but had no obvious growth promotion effect on cotton and maize. The colonization ability of PGPR in plant rhizosphere is a prerequisite for them to successfully promote plant growth [33]. The present study showed that strain NCD-2 had the strongest colonization ability in rhizosphere of eggplant; however, this strain showed a weaker colonization ability in rhizosphere of maize. Therefore, the rhizosphere colonization ability of this strain was partly related to its growth promotion effect. There were significant differences on growth promoting effects among crops after application of strain NCD-2. One of the possible reasons was that the optimal concentrations of strain NCD-2 application were differences among five crops. Therefore, it is necessary to further study the effects of application technique of strain NCD-2 on crop growth promotion. In addition, root exudation is one of the most important factors that affected bacterial colonization on the plant rhizosphere [34,35,36]. These different colonization results might be due to the different chemotaxis behaviors of strain NCD-2 responding to different plant exudates. The detailed mechanisms need to be studied further.

### 4.2. The Influence of Strain NCD-2 on Microbial Communities Varied among Plant Species

The rhizosphere microbiome plays an important role in regulating the nutrient cycle, plant growth and resistance to soil-borne plant diseases of crops. However, the structure of the rhizosphere microbiome is influenced by soil physical and chemical properties, plant species and even plant development stages [37]. Of these factors, plant species had the greatest influence on the structure of the rhizosphere microbiome [38,39]. In plant rhizospheres, the population density of microorganisms is as high as 109 CFUs/g soil, and with long-term evolution, plants and microorganisms form a relatively stable relationship [15]. Although the structure of the rhizosphere microbiome is influenced by environmental factors, the core microbiome is usually stable and plays a major role in affecting plant metabolism [40]. The results of this study showed that the rhizosphere microbiomes of five different crops were completely different but that the rhizosphere microbiomes of tomato, eggplant and pepper were more similar. Tomato, eggplant and pepper are all Solanaceae crops, and the rhizosphere microbiomes of these plant species are different from the rhizosphere microbiomes of cotton (*Malvaceae*) and maize (*Gramineae*). Therefore, the results of this study confirmed that the rhizosphere microbiome was plant species-dependent [38,39].

Changing the structure of the rhizosphere microbiome is one of the mechanisms by which PGPR promote crop growth [41]. Some studies have shown that microbial diversity can indicate microbial community stability [42,43], but some key or functional species are more important to plant growth than the overall microbial community diversity [44]. Microbial community can be detected by sequencing technology only when the population of specific microbial reached the detection threshold. Compared with CK, the abundances of microbial community changed after application of strain NCD-2, some microbial communities that did not recorded originally were detected, the same as some microbial communities with higher abundances dropped below the detection threshold due to the application of strain NCD-2. The colonization ability of beneficial microorganisms may be related to the active selection and/or targeted recruitment of specific microorganisms in plants. In the present study, there were significant differences on the abundances of beneficial microorganisms in rhizosphere soil of different crops after the application of strain NCD-2. The results showed that strain NCD-2 significantly recruited *Bacillus*, *Sphingobium*, *Fusarium*, *Clonostachys* and *Neurospora* in rhizosphere of tomato, increased *Lysobacter*, *Sphingobium*, *Pseudomonas*, *Byssochlamys*, *Penicillium*, *Mortierella* and *Cladorrhinum* in rhizosphere of eggplant, enriched *Lysobacter*, *Bacillus*, *Sphingobium*, *Byssochlamys*, *Acremonium* and *Peziza* in rhizosphere of pepper, increased *Lysobacter*, *Pseudomonas*, *Acremonium* and *Guehomyces* in rhizosphere of cotton, and recruited *Azotobacter*, *Conexibacter*, *Byssochlamys*, *Aspergillus*, *Trichoderma*, *Cephalotrichum* and *Neurospora* in rhizosphere of maize, respectively. Therefore, the microbial community was changed then to enter a new balance after application of strain NCD-2, and this change was the result of the interaction between strain NCD-2 and indigenous microbes. Similar results were observed in recent studies where microbial biocontrol agents’ treatment enriched other beneficial microorganisms in soil [20,21].

*Bacillus* is an important PGPR, and its plant growth-promoting mechanism includes the production of antibiotics to reduce the population of phytopathogens in the rhizosphere, the production of plant hormones and the dissolution of insoluble phosphorus in the soil into phosphorus, which is easily absorbed and utilized by plants [45,46,47]. *Sphingomonas* sp. is another important PGPR that can not only produce plant hormones such as auxin and gibberellin but also degrade toxic substances such as benzoin into benzamide, which is less toxic to plants, thus enhancing the stress tolerance of the plants and promoting plant growth [48,49]. In this study, the abundances of *Sphingomonas* were enriched in tomato, eggplant and pepper but decreased in the rhizosphere of maize after the application of strain NCD-2. *Lysobacter* was also enriched in eggplant, pepper and cotton. In addition, the abundances of *Mortierella* (in eggplant), *Fusarium* (in tomato), *Acremonium* (in pepper and cotton) and *Trichoderma* (in maize) were significantly increased after application of strain NCD-2. Many PGPR species, such as *Lysobacter* sp. [50], *Pseudomonas* sp. [51], *Trichoderma* [52,53] and *Mortierella* [54] are applied as biofungicides and biofertilizers to agricultural soils to enhance crop growth. Therefore, changing the abundances of the genera of beneficial microorganisms in the rhizosphere may be one of the reasons why strain NCD-2 promotes the growth of crops. *Fusarium* is an important soil-borne fungus that includes pathogenic fungi such as *Fusarium oxysporum*, which can cause *Fusarium* wilt in a variety of crops [55]. However, some *Fusarium* spp. can inhibit the growth of phytopathogens and show biological control effects [56,57]. The results of this study showed that the abundance of *Fusarium* in the rhizosphere of tomato was increased after application of strain NCD-2. Strain NCD-2 could inhibit the growth of many phytopathogens in vitro, through production of six antifungal active compounds [58]. However, strain NCD-2 showed weak or no inhibitory ability against the growth of *Fusarium*, *Trichoderma* and *Alternaria*, in contrast, *Fusarium* spp. could inhibit the growth of strain NCD-2 by production of fusaric acid (our unpublished data), and it was speculated that a large number of *Fusarium* in the rhizosphere could inhibit the growth of strain NCD-2, so lower colonization abilities of strain NCD-2 was observed in the tomato rhizosphere than in the eggplant rhizosphere in this study. In addition, root exudates may be changed after application of strain NCD-2, it will recruit different fungi. It may be the reason for the higher abundances of some fungi after being inoculated against strain NCD-2.

### 4.3. Variations in Microbial Functional Metabolic Genes for Five Crops by Strain NCD-2

Some studies showed microbial structural characteristics, metabolic pathways and gene regulation, to providing a comprehensive understanding of the relationships between the structure of microbial communities and their function [59,60]. Some researchers reported that *B. amyloliquefaciens* B1408 application significantly affected the function of the rhizosphere microbiome [22]. Other studies indicated that membrane transport in the soybean rhizosphere may be related to plant growth promotion [61]. The function of energy metabolism could enhance resistance to banana *Fusarium* wilt [62]. The functional metabolic genes observed could be elucidated by the relative abundance of a particular set of microbial categories in this study (Figure 6A, Appendix A). Functional prediction indicated diverse metabolic functions of the microbes among plant species after application of strain NCD-2. For instance, the relative abundances of amino acid transport and metabolism, nucleotide transport and metabolism, coenzyme transport and metabolism, lipid transport and metabolism, defense mechanisms were significantly enriched in rhizospheres of eggplant and pepper, while the relative abundances of above functional items were decreased in rhizospheres of tomato, cotton and maize, respectively. The relative abundance of transcription was significantly enriched in rhizosphere of tomato, the relative abundance of cell motility was significantly enriched in rhizosphere of cotton, and the relative abundance of cytoskeleton was enriched in rhizosphere of maize. The functions of secondary metabolites biosynthesis were enriched in the pepper and tomato rhizospheres after application of strain NCD-2. These results agreed with some scholars’ ideas that biosynthesis of secondary metabolites was increased in cucumber rhizosphere in the *B. amyloliquefaciens* B1408 treatment [22]. This observation could be caused by the application of strain NCD-2, providing a molecular base for microbial induced genetic enhancement for growth-promoting in different crops. These results also agreed with earlier studies that microbial agent can alter plant transcriptome [63]. Numerous studies had shown the ability of PGPR to activate a variety of defense responses in host plant tissue, in response to biotic and abiotic stress [64,65]. For instance, PGPR inoculation reduced disease severity and promoted growth through the induction of systemic defense in host tissue by improving the status of different defense enzymes [66]. In addition, some phenolics were accumulated by the inoculation of *Bacillus* species in pathogen pre-challenged conditions [67].

Variations in the composition of fungal functional groups inferred by FUNGuild showed that strain NCD-2 appeared to influence nutrition mode than CK treatment. For example, saprotrophs and symbiotrophs were abundant in eggplant and pepper rhizosphere after application of strain NCD-2. Analysis of the composition of fungal communities in the eggplant and pepper rhizospheres showed that strain NCD-2 treatment could simultaneously increase proportions of *Byssochlamys*, *Rhizopus*, *Aspergillus*, *Stachybotrys*, while the opposite tendencies were observed for the abundances of *Penicillium* and *Cephalotrichum*, respectively. The relative abundances of *Cephalotrichum*, *Penicillium*, *Preussia* and *Schizothecium* in cotton rhizosphere were enriched after application of strain NCD-2, while *Byssochlamys*, *Rhizopus*, *Trichoderma* and *Neurospora* were decreased, respectively. The relative abundances of *Byssochlamys*, *Trichoderma*, *Aspergillus*, *Cephalotrichum*, *Penicillium* and *Neurospora* in maize rhizosphere were enriched after application of strain NCD-2, while *Rhizopus*, *Stachybotrys*, *Melanospora*, *Staphylotrichum* and *Preussia* were decreased, respectively. The relative abundances of *Talaromyces*, *Neurospora*, *Stachybotrys*, *Melanospora* and *Staphylotrichum* were increased in tomato rhizosphere after application of strain NCD-2, while *Byssochlamys*, *Rhizopus*, *Trichoderma* and *Aspergillus* were decreased, respectively (Appendix A). The genus of *Byssochlamys* could produce bioactive compounds against plant pathogens [68]. *Penicillium* sp. is well known as a pathogen of several plant species causing fruit root [69]. *Schizothecium* sp. could improve the resistance to banana *Fusarium* wilt disease of the seedling [70]. *Trichoderma* sp. could control some fungal diseases in plant [71]. However, in our study, the relative abundances of some same genus were greatly differences in different plants rhizosphere after application of strain NCD-2 (Appendix A). Due to FUNGuild analysis was based on the literature and data, there were some limitations for analyze the functions to some extent. Thus, fungal functional groups need to be further studied.

## Figures and Tables

**Figure 1 microorganisms-11-00776-f001:**
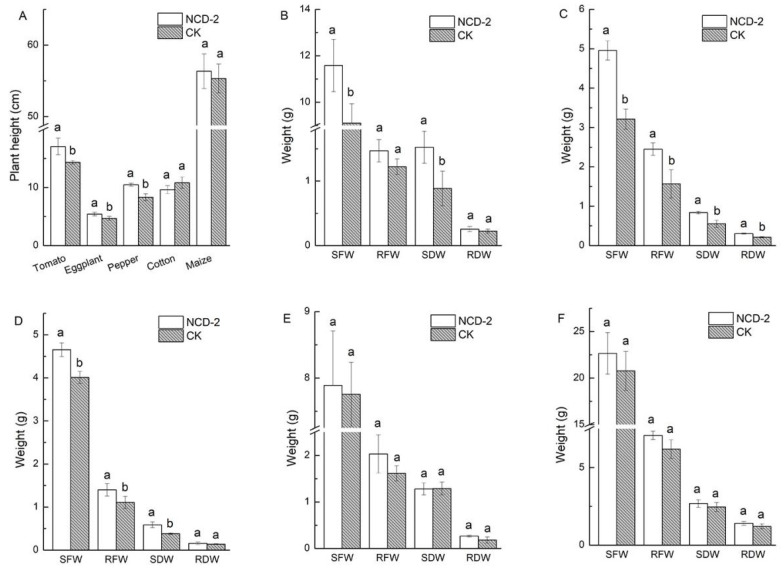
Effect of *Bacillus subtilis* NCD-2 on the growth indices of different crops. (**A**) Plant height; (**B**) plant fresh and dry weight of tomato; (**C**) plant fresh and dry weight of eggplant; (**D**) plant fresh and dry weight of pepper; (**E**) plant fresh and dry weight of cotton; (**F**) plant fresh and dry weight of maize. SFW represents shoot fresh weight, RFW represents root fresh weight, SDW represents shoot dry weight and RDW represents root dry weight. Each value is the mean of three replicates. Bars with the same lowercase letters indicate no significant difference between treatments based on Student’s *t*-test (*p* < 0.05). NCD-2 represents treatment with *B. subtilis* NCD-2, CK represents treatment with blank control.

**Figure 2 microorganisms-11-00776-f002:**
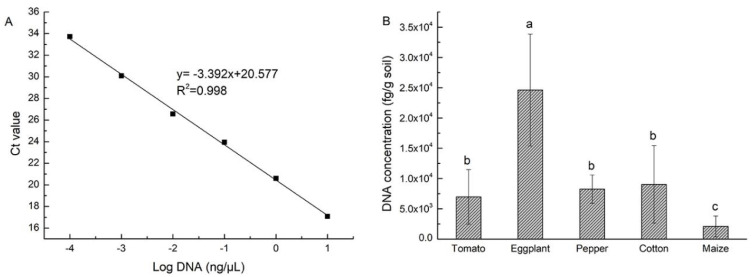
Quantitative detection of *Bacillus subtilis* strain NCD-2 in rhizosphere of different crops. (**A**) the establishment of standard curves; (**B**) colonization ability of strain NCD-2 in rhizosphere soil of different crops. Bars with the same lowercase letters indicate no significant difference between treatments based on one-way analysis of variance (ANOVA) (*p* < 0.05).

**Figure 3 microorganisms-11-00776-f003:**
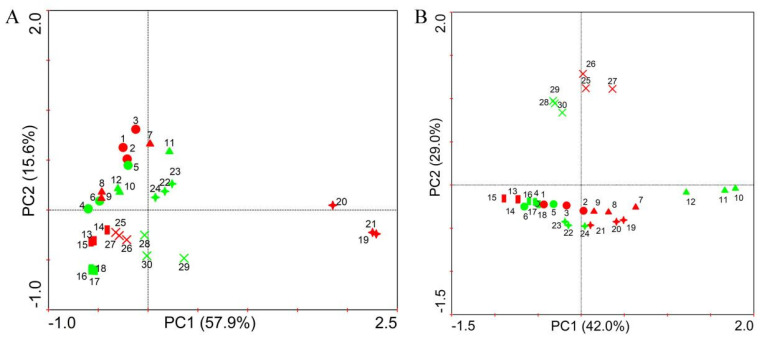
Principal component analysis of the soil bacterial (**A**) and fungal (**B**) communities collected from different treatments. 1–6 represent pepper, 7–12 represent tomato, 13–18 represent cotton, 19–24 represent eggplant, 25–30 represent maize. Red represents blank control (CK) treatment, green represents the treatment of strain NCD-2.

**Figure 4 microorganisms-11-00776-f004:**
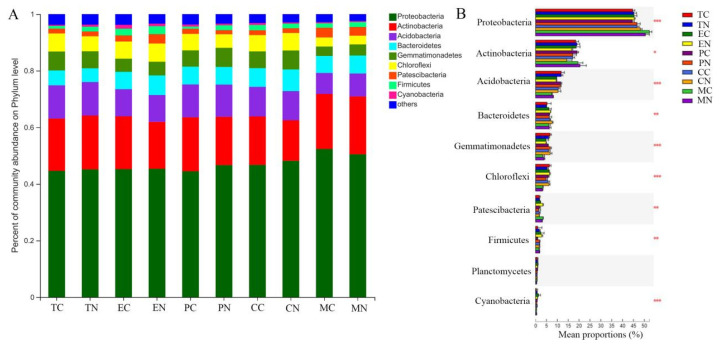
Analysis of compositions of bacterial communities from CK and NCD-2 treatments in different crops. (**A**) Percentage of community abundance at phylum level for each crop; (**B**) Significant analysis of community abundance at the phylum level. *, ** and *** means significantly at *p* < 0.05, *p* < 0.01 and *p* < 0.001 level, respectively. NCD-2 represents treatment with B. subtilis NCD-2, CK represents treatment with blank control. TC, tomato-CK; TN, tomato-NCD-2; EC, eggplant-CK; EN, eggplant-NCD-2; PC, pepper-CK; PN, pepper-NCD-2; CC, cotton-CK; CN, cotton-NCD-2; MC, maize-CK; MN, maize-NCD-2.

**Figure 5 microorganisms-11-00776-f005:**
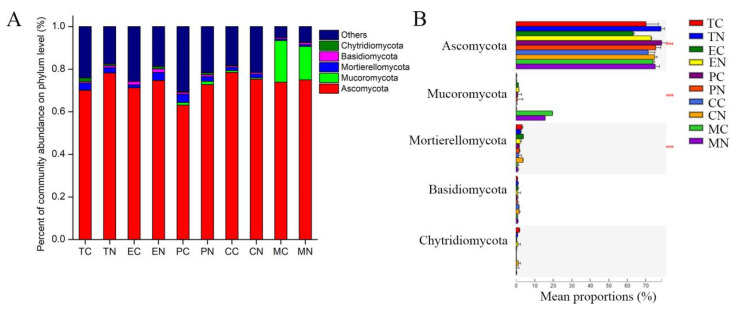
Analysis of compositions of fungal communities from CK and NCD-2 treatments in different crops. (**A**) Percentage of community abundance at phylum level for each crop; (**B**) Significant analysis of community abundance at the phylum level. *** means significantly at *p* < 0.001 level. NCD-2 represents treatment with B. subtilis NCD-2, CK represents treatment with blank control. TC, tomato-CK; TN, tomato-NCD-2; EC, eggplant-CK; EN, eggplant-NCD-2; PC, pepper-CK; PN, pepper-NCD-2; CC, cotton-CK; CN, cotton-NCD-2; MC, maize-CK; MN, maize-NCD-2.

**Figure 6 microorganisms-11-00776-f006:**
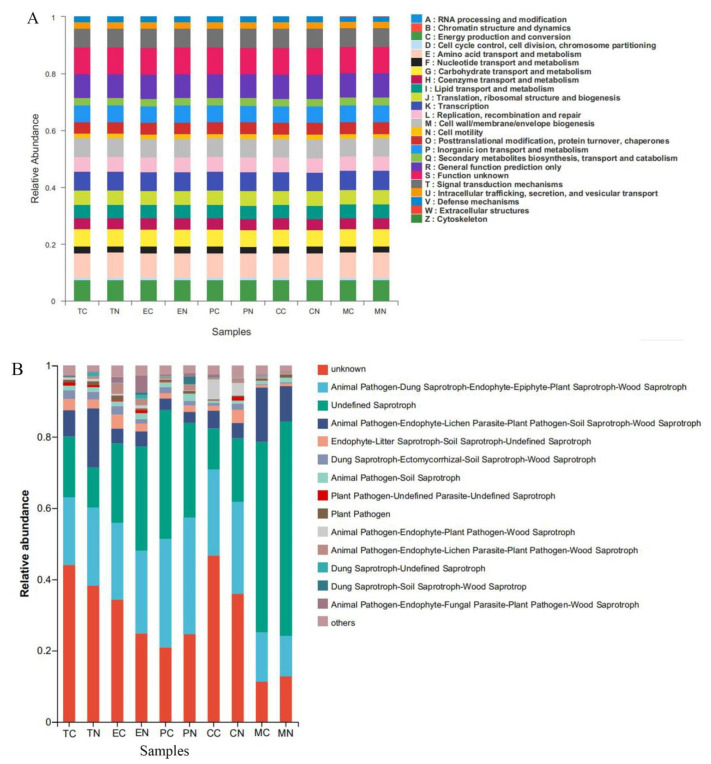
Predicted relative abundances of functional gene for bacteria and fungi by PICRUSt and FUNGild. (**A**) Bacteria; (**B**) Fungi. TC, tomato-CK; TN, tomato-NCD-2; EC, eggplant-CK; EN, eggplant-NCD-2; PC, pepper-CK; PN, pepper-NCD-2; CC, cotton-CK; CN, cotton-NCD-2; MC, maize-CK; MN, maize-NCD-2.

**Table 1 microorganisms-11-00776-t001:** ACE indices of bacteria and fungi in different crops after application of *B. subtilis* NCD-2.

Crop	Treatment	Bacteria	Fungi
Tomato	CK	5317.60 ± 110.67 a	752.42 ± 37.72 a
	NCD-2	5086.30 ± 115.61 a	648.89 ± 44.84 b
Eggplant	CK	5536.00 ± 357.83 a	787.24 ± 88.72 a
	NCD-2	5287.20 ± 109.10 a	684.16 ± 55.44 a
Pepper	CK	4996.70 ± 277.47 a	724.17 ± 97.82 a
	NCD-2	4922.60 ± 243.39 a	704.29 ± 18.04 a
Cotton	CK	5141.90 ± 309.11 a	565.92 ± 45.41 a
	NCD-2	4792.20 ± 129.30 a	531.79 ± 25.70 a
Maize	CK	4292.30 ± 440.07 a	506.52 ± 72.85 a
	NCD-2	4465.20 ± 45.48 a	561.22 ± 16.13 a

T; Data were expressed as the mean ± SE of three replicates. Means with the same letters are not significantly different according to Student’s *t*-test at *p* < 0.05. CK represents treatment with blank control, NCD-2 represents treatment with *B. subtilis* NCD-2.

## Data Availability

Raw data of 16S rRNA gene and ITS gene obtained from all samples are accessible via NCBI SRA database under accession numbers were PRJNA847255 and PRJNA847257, respectively.

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
