# Peer review of "Colonization Ability of Bacillus subtilis NCD-2 in Different Crops and Its Effect on Rhizosphere Microorganisms"

_microorganisms, 2023, doi:10.3390/microorganisms11030776_

Round 1

Reviewer 1 Report

The study "Colonization ability of Bacillus subtilis NCD-2 in different crops and its effect on rhizosphere microorganisms" investigated the colonization ability of the NCD-2 strain in tomato, eggplant, pepper, maize, and cotton, as well as the changes in rhizosphere bacterial and fungal community structure, and predicted the functional metabolic genes of rhizosphere microorganisms. The experimental design was reasonable, but the depth of data analysis and mining was insufficient. Overall, the study has certain academic value and practical significance, but lacks some innovation. It is recommended to revise and resubmit the article.

The following are some issues with the article:

1. The summary of the mechanism by which the NCD-2 strain promotes plant growth is not clear enough.

2. The analysis of rhizosphere bacterial and fungal community classification is too complicated and lacks emphasis.

3. The discussion of the experimental results is not sufficiently in-depth. For example, the correlation between the colonization ability of the NCD-2 strain and its growth-promoting effect in 4.1; the discussion of the functional metabolic genes of microorganisms and the functional of fungal communities should be further refined.

Author Response

Dear Editor and Reviewers,

Thank you very much for your kind help and the reviewers’ comments on our manuscript entitled “Colonization ability of Bacillus subtilis NCD-2 in different crops and its effect on rhizosphere microorganisms” (Manuscript Number: 2263622). We provide this cover letter to explain, point by point, the details of our revisions in the manuscript and our responses to the reviewers’ comments. We hope the revised manuscript would meet the requirement of the Journal and satisfy the reviewers.

Looking forward to hearing from you soon.

Best Regards,

Ping MA

1. The summary of the mechanism by which the NCD-2 strain promotes plant growth is not clear enough.

Reply: Thank you for your suggestion. It has been revised at the corresponding position in the manuscript. The following sentence was added in line 25-27.

Based on the results obtained in this study, it was concluded that the growth promoting ability of strain NCD-2 were correlated with its colonization quantity and the microbial species it recruited.

 2. The analysis of rhizosphere bacterial and fungal community classification is too complicated and lacks emphasis.

Reply: Thank you for your suggestion. It has been revised at the corresponding position in the manuscript.

In terms of bacterial community, the following sentences were revised in line 282-288, 291-303.

After application of strain NCD-2, the relative abundances of Proteobacteria and Firmicutes in tomato, eggplant and pepper rhizospheres were more abundant than CK treatment, while the relative abundances of Actinobacteria, Acidobacteria and Cyanobacteria were decreased in eggplant and pepper. The relative abundances of Bacteroidetes and Gemmatimonadetes in cotton and maize rhizosphere were more abundant than CK treatment, while Patescibacteria and Cyanobacteria were decreased.

At the genus level, there were significant differences in dominant genera abundances of different crops between CK and NCD-2 treatments based on Student’s t-test (P < 0.05)(Table S3). Bacillus and Sphingobium were significantly higher abundances in the rhizospheres of tomato, eggplant and pepper under application of strain NCD-2 than that of CK treatment, only Lysobacter was significantly higher abundance in the rhizospheres of eggplant and pepper, Ramlibacter was significantly higher abundance in the rhizospheres of tomato and eggplant. Lysobacter, Bdellovibrio, Pseudomonas, Peredibacter, Limnobacter and Pedobacter were more abundances in the cotton rhizosphere after application of strain NCD-2, while lower abundances of MND1, Kribbella, Rhodococcus and Actinoplanes. Azotobacter and Conexibacter were more abundances in the maize rhizosphere after application of strain NCD-2, while Lysobacter, Caulobacter, Brevundimonas, Acidovorax, Kaistia, Pseudoxanthomonas, Kribbella and Sphingobium were lower abundances.

In terms of fungal community, the following sentence was revised in line 319-321, 323-338.

Ascomycota and Basidiomycota were more abundant in rhizospheres of different crops after application of strain NCD-2 than CK treatment. Analysis of variance showed that the differences in the mean proportions of Ascomycota, Mucoromycota, Mortierellomycota in different crops were significant (Figure 5B).

At the genus level, there were significant differences in dominant genera abundances of different crops between NCD-2 and CK treatments based on Student’s t-test (P < 0.05)(Table S4). For tomato, Fusarium, Clonostachys and Neurospora in the rhizosphere were significantly higher abundances after application of strain NCD-2 than CK treatment, while the lower abundances of Byssochlamys, Aspergillus and Fibulochlamys. For eggplant, the higher abundances of Byssochlamys, Penicillium, Mortierella and Cladorrhinum in the rhizosphere after application of strain NCD-2, while Fibulochlamys was significantly decreased. For pepper, Byssochlamys, Acremonium and Peziza in the rhizosphere were significantly higher abundances under application of strain NCD-2, while lower abundances of Verticillium, Nectria and Guehomyces. For cotton, the abundance of Acremonium and Guehomyces in the rhizosphere were significantly increased after application of strain NCD-2, while Byssochlamys, Neurospora and Arthrobotrys were decreased. For maize, Byssochlamys, Aspergillus, Trichoderma, Cephalotrichum and Neurospora in the rhizosphere were higher abundances after application of strain NCD-2, while lower abundances of Arthrobotrys, Peziza and Simplicillium.

3. The discussion of the experimental results is not sufficiently in-depth. For example, the correlation between the colonization ability of the NCD-2 strain and its growth-promoting effect in 4.1; the discussion of the functional metabolic genes of microorganisms and the functional of fungal communities should be further refined.

Reply: Thank you for your suggestion. It has been revised at the corresponding position in the manuscript.

As for the correlation between the colonization ability of the NCD-2 strain and its growth-promoting effect in 4.1,

the following sentence was revised in line 418-421.

For instance, the colonization pattern of Bacillus subtilis N11, which forming biofilms along the elongation and differentiation zones of banana roots, could be linked to the mechanism of growth-promoting effect and biological control [32].

The following sentence and references were added in line 429-433.

In addition, root exudation is one of the most important factors that affected bacterial colonization on the plant rhizosphere [34-36]. These different colonization results might be due to the different chemotaxis behaviors of strain NCD-2 responding to different plant exudates. The detailed mechanisms need to be studied further.

[34] Wen, T.; Zhao, M.L.; Yuan, J.; Kowalchuk, G.; Shen, Q.R. Root exudates mediate plant defense against foliar pathogens by recruiting beneficial microbes. Soil Ecol. Lett.2021, 3, 42-51. 

[35] Singh,B.K.; Trivedi, P.; Egidi, E.; Macdonald, C.A.; Baquerizo, M.D. Crop microbiome and sustainable agriculture. Nat. Rev. Microbiol2020,18, 601-602.

[36] Stephen, A.R.; Joseph, G.; Jurriaan, T. Crying out for help with root exudates: adaptive mechanisms by which stressed plants assemble health-promoting soil microbiomes. Opin. Microbiol. 2019, 49, 73-82.

The following sentences were deleted in revised manuscript.

Strain NCD-2 significantly stimulated the growth of different crops (such as tomato, eggplant and pepper), including plant height, shoot and root fresh (dry) weight were increased to different degrees compared to the CK treatment in the pot experiment. Although there was no significantly influence on growth for cotton and maize, plant growth indices roughly increased following strain NCD-2 application (Figure 1).

The following sentences were revised in line 513-519, and some references were added.  

Some studies showed microbial structural characteristics, metabolic pathways, and gene regulation, to providing a comprehensive understanding of the relationships between the structure of microbial communities and their function [59,60]. Some researchers reported that B. amyloliquefaciens B1408 application significantly affected the function of the rhizosphere microbiome [22]. Other studies indicated that membrane transport in the soybean rhizosphere might be related to plant growth promotion [61]. The function of energy metabolism could enhance resistance to banana Fusarium wilt [62].

[59] Ogundeji, A.O.; Li, Y.; Liu, X.J.; Meng, L.B.; Sang, P.; Mu, Y.; Wu, H.L.; Ma, Z.N.; Hou, J.; Li, S.M. Eggplant by grafting enhanced with biochar recruits specific microbes for disease suppression of Verticillium wilt. Appl. Soil Ecol. 2021, 163, 103912.

[60] Dini-Andreote, F; van Elsas, J.D.; Olff, H.; Salles, J.F. Dispersal-competition tradeoff in microbiomes in the quest for land colonization. Sci. Rep. 2018, 8, 9451.

[61] Berg, N.V.D.; Berger, D.K.; Hein, I.; Birch, P.R.J.; Wingfield, M.J.; Viljoen, A. Tolerance in banana to Fusarium wilt is associated with early up-regulation of cell wall-strengthening genes in the roots. Mol. Plant Pathol. 2007, 8, 333-341.

[62] Mendes, L.W.; Kuramae, E.E.; Navarrete, A.A.; van Veen, J.A.; Tsai, S.M. Taxonomical and functional microbial community selection in soybean rhizosphere. ISME J. 2014, 8, 1577-1587.

The following sentences were revised in line 523-535.

For instance, the relative abundances of amino acid transport and metabolism, nucleotide transport and metabolism, coenzyme transport and metabolism, lipid transport and metabolism, defense mechanisms were significantly enriched in rhizospheres of eggplant and pepper, while the relative abundances of above functional items were decreased in rhizospheres of tomato, cotton and maize, respectively. The relative abundance of transcription was significantly enriched in rhizosphere of tomato, the relative abundance of cell motility was significantly enriched in rhizosphere of cotton, and the relative abundance of cytoskeleton was enriched in rhizosphere of maize. The functions of secondary metabolites biosynthesis were enriched in the pepper and tomato rhizospheres after application of strain NCD-2. These results agreed with some scholars’ ideas that biosynthesis of secondary metabolites were increased in cucumber rhizosphere in the B. amyloliquefaciens B1408 treatment [22].

The following sentences were revised in line 548-570.

Analysis of the composition of fungal communities in the eggplant and pepper rhizospheres showed that strain NCD-2 treatment could simultaneously increase proportions of Byssochlamys, Rhizopus, Aspergillus, Stachybotrys, while the opposite tendencies were observed for the abundances of Penicillium and Cephalotrichum, respectively. The relative abundances of Cephalotrichum, Penicillium, Preussia and Schizothecium in the cotton rhizosphere were enriched after application of strain NCD-2, while Byssochlamys, Rhizopus, Trichoderma, and Neurospora were decreased, respectively. The relaitve abundances of Byssochlamys, Trichoderma, Aspergillus, Cephalotrichum, Penicillium and Neurospora in maize rhizosphere were enriched after application of strain NCD-2, while Rhizopus, Stachybotrys, Melanospora, Staphylotrichum and Preussia were decreased, respectively. The relative abundances of Talaromyces, Neurospora, Stachybotrys, Melanospora and Staphylotrichum were increased in tomato rhizosphere after application of strain NCD-2, while Byssochlamys, Rhizopus, Trichoderma and Aspergillus were decreased, respectively (Table S4). The genus of Byssochlamys could produce bioactive compounds against plant pathogens [68]. Penicillium sp. is well known as a pathogen of several plant species causing fruit root [69]. Schizothecium sp. could improve the resistance to banana Fusarium wilt disease of the seedling [70]. Trichoderma sp. could control some fungal diseases in plant [71]. However, in our study, the relative abundances of some same genus were greatly differences in different plants rhizosphere after application of strain NCD-2 (Table S4). Due to FUNGuild analysis was based on literature and data, there were some limitations for analyze the functions to some extent. Thus, fungal functional groups should be needed to further studied.

[68] Rodrigo, S.; Santamaria, O.; Halecker, S.; Lledó, S.; Stadler, M. Antagonism between Byssochlamys spectabilis (anamorph Paecilomyces variotii ) and plant pathogens: Involvement of the bioactive compounds produced by the endophyte. Ann. Appl. Biol. 2017, 171(3), 464-476.

[69] Liu, Y.; Wang, W.H.; Zhou, Y.H.; Yao, S.X.; Deng, L.L.; Zeng, K.F. Isolation, identification and in vitro, screening of chongqing orangery yeasts for the biocontrol of Penicillium digitatum, on citrus fruit. Biol. Control 2017, 110, 18–24.

[70] Nong, Q.; Zhang,  W.L.; Lan, T.j.; Su, Q.; Chen, Y.L.; Zhang, Y.; Qin, L.P.; Xie, L. Screening and identification of dark septate endophyte strain L-14 and its mechanism of banana Fusariumwilt Disease Resistance. Chinese Journal of Tropical Crops, 2017, 38, 559-564.

[71] Sánchez-Montesinos, B.; Santos, M.; Moreno-Gavíra, A.; Marín-Rodulfo, T.; Gea, F.J.; Diánez, F. Biological control of fungal diseases by Trichoderma aggressivum f. europaeum and Its compatibility with fungicides. Journal of Fungi, 2021, 7(8), 598.

Reviewer 2 Report

The manuscript describes effect of the local isolate of Bacillus subtilis on rhizosphere bacteria and fungi on five different crops.

Figure 1: Add what SFW, RFW, SDW and RDW stand for in the legend. 

Line 205: Change "Quantitatively detected for the quantity of" to "Quantitative detection of Bacillus subtilis."  

Line 231: Add tomato before eggplant.

Table 1: Provide the full description for "ACE."

Line 240: Provide the full description for "PCA" here.

Discuss the possible reason why predicted relative abundances of functional gene for bacteria are the same throughout the five crops and with or without treatment.

Author Response

Dear Editor and Reviewers,

Thank you very much for your kind help and the reviewers’ comments on our manuscript entitled “Colonization ability of Bacillus subtilis NCD-2 in different crops and its effect on rhizosphere microorganisms” (Manuscript Number: 2263622). We provide this cover letter to explain, point by point, the details of our revisions in the manuscript and our responses to the reviewers’ comments. We hope the revised manuscript would meet the requirement of the Journal and satisfy the reviewers.

Looking forward to hearing from you soon.

Best Regards,

Ping MA

Figure 1: Add what SFW, RFW, SDW and RDW stand for in the legend.

Reply: Thank you for your suggestion. It has been revised at the corresponding position in the manuscript. The following sentence was revised in line 195-196.

SFW represents shoot fresh weight, RFW represents root fresh weight, SDW represents shoot dry weight, and RDW represents root dry weight.

Line 205: Change "Quantitatively detected for the quantity of" to "Quantitative detection of Bacillus subtilis."  

Reply: Thank you for your suggestion. It has been revised at the corresponding position in the manuscript. The following sentence was revised in line 214.

Quantitative detection of Bacillus subtilis strain NCD-2 in rhizosphere of different crops.

Line 231: Add tomato before eggplant.

Reply: Thank you for your question.

The following sentence was revised in line 239-243.

For fungi, ACE index was 648.89±44.84 in tomato rhizosphere after application of strain NCD-2, there was significantly increased in CK treatment (752.42±37.72). Although ACE indices were lower in the rhizospheres of eggplant, pepper and cotton after application of strain NCD-2 than that of CK treatment, there were no significant differences.

Table 1: Provide the full description for "ACE."

Reply: Thank you for your suggestion.

ACE is an indicator of microbial diversity index. To be specific, ACE represents the parameter of the richness index. ACE is one of many indices that represent the microbial diversity index. ACE is a generic expression, like DNA, RNA, PCR and so on.

Line 240: Provide the full description for "PCA" here.

Reply: Thank you for your suggestion. It has been revised at the corresponding position in the manuscript.

Discuss the possible reason why predicted relative abundances of functional gene for bacteria are the same throughout the five crops and with or without treatment.

 Rely: Thanks for your question. It has been revised at the corresponding position in the manuscript.

The following sentences were revised in line 513-519, and some references were added.  

Some studies showed microbial structural characteristics, metabolic pathways, and gene regulation, to provide a comprehensive understanding of the relationships between the structure of microbial communities and their function [59,60]. Some researchers reported that B. amyloliquefaciens B1408 application significantly affected the function of the rhizosphere microbiome [22]. Other studies have indicated that membrane transport in the soybean rhizosphere may be related to plant growth promotion [61]. The function of energy metabolism could enhance resistance to banana Fusarium wilt [62].

[59] Ogundeji, A.O.; Li, Y.; Liu, X.J.; Meng, L.B.; Sang, P.; Mu, Y.; Wu, H.L.; Ma, Z.N.; Hou, J.; Li, S.M. Eggplant by grafting enhanced with biochar recruits specific microbes for disease suppression of Verticillium wilt. Appl. Soil Ecol. 2021, 163, 103912.

[60] Dini-Andreote, F; van Elsas, J.D.; Olff, H.; Salles, J.F. Dispersal-competition tradeoff in microbiomes in the quest for land colonization. Sci. Rep. 2018, 8, 9451.

[61] Berg, N.V.D.; Berger, D.K.; Hein, I.; Birch, P.R.J.; Wingfield, M.J.; Viljoen, A. Tolerance in banana to Fusarium wilt is associated with early up-regulation of cell wall-strengthening genes in the roots. Mol. Plant Pathol. 2007, 8, 333-341.

[62] Mendes, L.W.; Kuramae, E.E.; Navarrete, A.A.; van Veen, J.A.; Tsai, S.M. Taxonomical and functional microbial community selection in soybean rhizosphere. ISME J. 2014, 8, 1577-1587.

The following sentences were revised in line 523-535.

For instance, the relative abundances of amino acid transport and metabolism, nucleotide transport and metabolism, coenzyme transport and metabolism, lipid transport and metabolism, defense mechanisms were significantly enriched in rhizospheres of eggplant and pepper, while the relative abundances of above functional items were decreased in rhizospheres of tomato, cotton and maize, respectively. The relative abundance of transcription was significantly enriched in rhizosphere of tomato, the relative abundance of cell motility was significantly enriched in rhizosphere of cotton, and the relative abundance of cytoskeleton was enriched in rhizosphere of maize. The functions of secondary metabolites biosynthesis were enriched in the pepper and tomato rhizospheres after application of strain NCD-2. These results agreed with some scholars’ ideas that biosynthesis of secondary metabolites were increased in cucumber rhizosphere in the B. amyloliquefaciens B1408 treatment [22].
